# Comparing Cytology Brushes for Optimal Human Nasal Epithelial Cell Collection: Implications for Airway Disease Diagnosis and Research

**DOI:** 10.3390/jpm13050864

**Published:** 2023-05-21

**Authors:** Laura K. Fawcett, Nihan Turgutoglu, Katelin M. Allan, Yvonne Belessis, John Widger, Adam Jaffe, Shafagh A. Waters

**Affiliations:** 1Discipline of Paediatrics and Child Health, School of Clinical Medicine, Faculty of Medicine and Health, UNSW Sydney, Sydney, NSW 2052, Australia; laura.fawcett@unsw.edu.au (L.K.F.); n.turgutoglu@unsw.edu.au (N.T.); katelin.allan@student.unsw.edu.au (K.M.A.); y.belessis@unsw.edu.au (Y.B.); john.widger@sa.gov.au (J.W.); a.jaffe@unsw.edu.au (A.J.); 2Department of Respiratory Medicine, Sydney Children’s Hospital, Sydney, NSW 2031, Australia; 3School of Biomedical Sciences, Faculty of Medicine and Health, UNSW Sydney, Sydney, NSW 2052, Australia

**Keywords:** nasal epithelial cells, cytology brush, nasal brushing, cilia, ciliary beat frequency, primary cell culture, PCD, cystic fibrosis, airway disease, children

## Abstract

Primary nasal epithelial cells and culture models are used as important diagnostic, research and drug development tools for several airway diseases. Various instruments have been used for the collection of human nasal epithelial (HNE) cells but no global consensus yet exists regarding the optimal tool. This study compares the efficiency of two cytology brushes (Olympus (2 mm diameter) and Endoscan (8 mm diameter)) in collecting HNE cells. The study involved two phases, with phase one comparing the yield, morphology and cilia beat frequency (CBF) of cells collected from paediatric participants using each of the two brushes. Phase two compared nasal brushing under general anaesthetic and in the awake state, across a wide age range, via the retrospective audit of the use of the Endoscan brush in 145 participants. Results indicated no significant difference in CBF measurements between the two brushes, suggesting that the choice of brush does not compromise diagnostic accuracy. However, the Endoscan brush collected significantly more total and live cells than the Olympus brush, making it a more efficient option. Importantly, the Endoscan brush is more cost-effective, with a notable price difference between the two brushes.

## 1. Introduction

The structure and function of the airway epithelium plays a crucial role in the development and progression of airway disease. To aid in the diagnosis and study of such disease, a versatile technique known as nasal brushing can be performed safely on individuals of all ages, including healthy, non-sedated neonates [1,2,3]. The procedure collects human nasal epithelial (HNE) cells from the inferior nasal turbinate, which are particularly useful in diagnosing primary ciliary dyskinesia (PCD) and are increasingly used in research of other airway diseases [2,4,5,6,7,8,9]. The accuracy and reliability of PCD diagnoses are directly influenced by the yield of epithelial cells collected from nasal brushings. Sufficient cell yield is necessary to conduct adequate assessment of ciliary beat frequency (CBF) and coordination, as well as the assessment of ultrastructural defects and protein distribution anomalies in cilia via electron microscopy and immunofluorescence (IF) staining.

The cell yield also influences the success of HNE cell cultures, which are a robust and informative diagnostic aid in distinguishing primary from secondary abnormalities of cilia [10]. The reanalysis of cilia is often necessary following HNE cell culture and differentiation at the air–liquid interface to investigate the potential impact of secondary infections on CBF, which can modify ciliary function and lead to the potential misinterpretation of results in PCD assessment. Beyond PCD, HNE cell cultures have become increasingly important for diagnosis and studying the pathophysiology of, and developing drugs for, other airway diseases, such as allergic rhinitis, asthma and cystic fibrosis (CF) [9,11,12,13,14,15,16].

Various instruments, such as swabs [17], curettes [3], interdental [2], cytology [18] and custom-made brushes [9], have been used for HNE cell sampling. Prior studies have identified a modified interdental brush to be the most efficient in collecting HNE cells in comparison to curettes or swabs [19]. As demand for HNE cell collection in both clinical and research settings grows, it is increasingly important to identify an instrument that is small enough to fit through the nares, minimises pain and trauma, collects a high number of viable cells and is cost-effective. To ensure sufficient cell yield, it is essential to adhere to standardised protocols for nasal brushing, sample handling and processing. When the initial sample does not provide enough cells for assessment, a repeat nasal brushing may be necessary. While there is limited consensus on the ideal collection tool, further standardisation and optimisation of the procedure are necessary.

To address the need for an optimal HNE cell collection instrument, we conducted a study to compare two Therapeutic Goods Administration (Australia)-approved cytology brushes (Figure 1). Our study aims to provide a stronger scientific basis for selecting the most suitable instrument for HNE cell collection in both clinical and research settings by comparing these two brushes.

## 2. Materials and Methods

### 2.1. Participants

All participants were enrolled in the CF Avatar study, which was approved by the Sydney Children’s Hospital Network Human Research Ethics Committee (HREC/16/SCHN/120). All participants and/or their parents/guardians signed a written consent form for their participation. Paediatric patients were recruited through attendance at Sydney Children’s Hospital CF clinic or referral for a bronchoscopy. Adult patients were recruited from the Royal Prince Alfred CF clinic.

### 2.2. Cytology Brushes

Two different brushes were used to sample nasal epithelial cells from the inferior nasal turbinates. One, a small cytology brush (BC-202D-2010, Olympus, Mount Waverley, VIC, Australia), which is commonly used to collect cells from the lower airway epithelium is used to brush the nasal turbinate in clinical PCD diagnostic laboratories in Australia. The second brush, a larger cytology brush (Endoscan, 33009-SA McFarlane, Ringwood, VIC, Australia), is typically used to collect cells from the cervical mucosa for cervical cell cytology examination. The Endoscan cytology brush cost less than AUD 0.50 while the Olympus brush had a price of AUD 42.50 (2019/2020 costs) (Figure 2). 

### 2.3. Collection of Cells from Nasal Inferior Turbinates of Participants

We followed our published protocol for nasal brushing [20]. Briefly, in phase one of the study, in order to compare the two brushes, a group of paediatric participants (Table 1; *n* = 13) underwent paired nasal brushing while under general anaesthetic for a clinically indicated procedure. They were supine with their head in the “sniffing” position to maintain a patent airway. We applied 0.9% saline drops to the nasal passage and removed visible mucus with gauze prior to brushing the inferior nasal turbinate. Each participant underwent the brushing of one inferior nasal turbinate with the Olympus brush and brushing of the opposite inferior nasal turbinate with the Endoscan brush. The brushing procedure was controlled, with the same operator carrying out all procedures and performing three rotations with each brush in each nostril. This was possible to perform without discomfort due to the anaesthetised state of the patient. In phase two, a group of participants were brushed while under general anaesthesia (GA) or awake (Table 2; *n* = 145). Awake participants blew their nose to remove mucous prior to the nasal brushing and at least one nostril was brushed with one Endoscan brush. Children under the age of five were not approached for an awake nasal brushing, as the CF clinic routinely performs annual bronchoscopy on this age group.

### 2.4. Cell Count

Briefly, the collected HNE cells were dislodged from the brush and dissociated by gentle vortexing and passage through a cell strainer (Sigma CLS431750), prior to pelleting and reconstitution in 1 mL of media, as previously described [20]. Cell counts were performed using an automated system (Countess II Automated cell counter, ThermoFisher Scientific, Waltham, MA, USA), according to the manufacturer’s instructions. Total and live cell counts were recorded.

### 2.5. Cell Culture

We employed a feeder-free protocol to culture the collected HNE cells, a widely accepted method for airway cell culture [21,22]. Briefly, cells were seeded in a collagen-I coated six-well plate. For comparison of the Olympus and Endoscan brushes, an equal cell count was used from each brush. The culture media used was Bronchial Epithelial Cell Medium (ScienCell Research Laboratories 3211) supplemented with 1 μM A83-01 (Tocris Bioscience 2939), 1 μM DMH1 (Selleckchem S7146), 3.3 nM EC23 (Enzo Life Sciences BML-EC23-0500) and 10 μM Y-27632 [22]. An antimicrobial cocktail of gentamicin, penicillin—streptomycin and Amphotericin B was added, as per Allan et al. [20]. The medium was changed every second day until the culture reached confluency (approximately 20 days post seeding). Cultures were then imaged on an EVOS fl digital microscope (AMF-4303, AMG). Following this, cells were dissociated with trypsin/EDTA (Lonza CC-5034) for 5–7 min at 37 °C and neutralized with trypsin neutralizing solution (Lonza CC-5034). Cell counts were performed as above. The population doubling rate was calculated using the following formula $Population\;doubling\;rate\;Number\;of\;days\;in\;culture\times \frac{LOG\left(2\right)}{LOG\left(End\;cell\;count\right)-LOG\left(Seeding\;cell\;count\right)}$ taken from the ATCC animal cell culture guide [22].

### 2.6. Ciliary Beat Frequency Measurements

Ciliary beat frequency (CBF) was measured from nasal epithelial sheets collected from each brush as per our previously published protocol [20]. Briefly, collected nasal epithelial sheets were imaged using a high-speed live cell imaging system (Eclipse Ti2-E, Nikon microscope with an Andor Zyla 4.2 sCMOS camera and a CFI S Plan Fluor ELWD 20×/0.45 objective) in an environmentally controlled chamber (37 °C, 85% humidity and 5% CO_2_) on the same day that they were acquired from the participant [20]. Time-lapse images were acquired and CBF analysed using a custom-built script in Matlab v9.0.0 (MathWorks, Natick, MA, USA) [20].

### 2.7. Statistical Analysis

In phase one, for cell count comparisons, a Wilcoxon matched pairs signed-rank test was performed. In phase two, for cell count comparisons, log transformation was applied before performing a *t*-test. A *p*-value of less than 0.05 was considered statistically significant. All statistical analysis was performed using GraphPad Prism v9.5.1 (GraphPad Software, San Diego, CA, USA). UNSW Stats Central were consulted regarding the statistical analysis for this study.

## 3. Results

### 3.1. Enhanced Cell Total and Viability with Endoscan Cytology Brush Selection

Our findings revealed a significant difference in cell counts obtained using the Olympus or Endoscan cytology brushes when brushing the participants’ inferior nasal turbinates (Appendix A; *n* = 7). The total cell count was higher for cells acquired using the Endoscan brush compared to the Olympus brush in six out of seven participants (Figure 3A; Differences ranged from −0.4 × 10^6^ to +4.7 × 10^6^; *p* = 0.031). Notably, the live cell count was greater for the cells collected using the Endoscan brush as opposed to the Olympus brush in all seven participants (Figure 3B; differences ranged from 0.5 × 10^5^ to 6.8 × 10^5^; *p* = 0.016). The culture of the cells, at equal live cell seeding density, verified that live cells maintained their expansion capacity and cobblestone cell morphology, which is consistent with airway epithelial cell morphology (Figure 3C and Appendix A).

### 3.2. Ciliary Beat Frequency Measurements Were Not Impacted by Cytology Brush Choice

To explore whether the utilisation of two distinct cytology brushes would impact CBF measurements, participants (*n* = 4) underwent paired nasal brushings with both brushes. CBF measurements for freshly brushed (as opposed to cultured) nasal epithelial cell sheets acquired using both brushes were compared. All CBF measurements fell within the expected physiological range [23]. No difference was found between the CBF measurements of nasal epithelial sheets acquired using the two brushes, with results within 1.5 Hz of the other (range −1.3 to +1.5 Hz) (Figure 4). These findings suggest that the choice of cytology brush does not compromise the accuracy of the CBF-based diagnosis.

### 3.3. Effectiveness of Endoscan Brush for Nasal Brushing in Awake Cystic Fibrosis Patients

To control for participant cooperation when comparing the Olympus and Endoscan brushes, phase one of the study only involved brushing children under general anaesthesia. However, in clinical settings, awake nasal brushing is the preferred approach as it eliminates the risk associated with general anaesthesia. Therefore, we aimed to compare the efficacy of the nasal brushing technique in anaesthetised versus awake participants. Having established that the larger Endoscan brush was able to collect more cells without a negative impact on cell culture and CBF, we used this brush to collect cells for phase two of our research study, which aimed to establish a biobank of airway cells (miCF-AVATAR Biobank).

To ensure comprehensive evaluation across various age groups, a total of 145 participants with CF, comprising 52 anaesthetised children and adolescents (6 months–17 years) and 93 awake children, adolescents and adults (5–58 years) underwent nasal brushings between 2017 and 2019 (Table 2).

Our findings revealed that significantly more total and live cells are collected from patients when under GA (Figure 5, Appendix A; 4.9 × 10^6^ vs. 1.4 × 10^6^; *p* < 0.001, 1.7 × 10^6^ vs. 0.4 × 10^6^; *p* < 0.001). It is important to consider that the operator may brush differently when participants are under general anaesthesia, which could contribute to the observed differences in cell collection. Our findings support the use of nasal brushing as a minimally invasive technique for collecting nasal epithelial cells. Whilst lower than the cell counts achieved from patients under GA, cell counts obtained from the awake participants were considered acceptable seeding cell counts for cell culture and expansion. A retrospective audit of the cell culture records revealed that 130 of the 145 participants had successful cultures that were cryopreserved, indicating a culture success rate of 89.6%. Two out of fifteen participants with failed cultures had cell counts of less than 160,000 (1.7 × 10^4^ and 1.6 × 10^5^). The remaining 13 participants’ collected cell counts ranged from 1.9 × 10^5^–8.0 × 10^6^, but the cultures failed due to resistant microbial contamination.

## 4. Discussion

HNE cells play an important role in various cell culture techniques and in investigating patients for PCD and other respiratory diseases [1,2,3,4,5,6,7,8,9,10,11,12]. To our knowledge, this is the first comparison of two different cytology brushes for the purpose of collecting nasal epithelial cells for cell culture and CBF measurement. We found no significant difference in the CBF measurement or cell culture phenotype between the samples obtained from the same participant via different brushes, suggesting that brush choice did not compromise the accuracy of CBF measurement.

The Endoscan cytology brush collected more cells than the Olympus brush for the same participant. This included a higher number of both total cells and live cells. When comparing the two brushes, the brushing procedure in our study was carried out by a single operator while the participants were under GA, ensuring a consistent sampling environment and minimising participant cooperation from being a confounding factor. The cell counter used in this study identifies all cell types, including blood cells, should they be present in the collected sample. As such, traumatic brushing could lead to elevated cell counts. However, no significant trauma was recorded for any participant in our study in their medical records. Although the Endoscan brush has a larger diameter than the Olympus brush, it features softer bristles that cover the entire brush area. This larger surface area may allow a more efficient collection of cells from the inferior nasal turbinates.

The Endoscan brush, with an 8 mm diameter, was effectively utilised for brushing the nasal passage in children as young as six months old under GA. This finding indicates that the size of the Endoscan brush is not a limiting factor when it comes to the size of the nares and can be appropriately used even in young paediatric patients. This is despite it being larger than the 2.7 mm diameter interdental brush used in a previous study by Miller et al. with an infant cohort [1]. Cell counts were not reported by Miller et al., but they successfully cultured cells from nasal brushings obtained from neonates who were less than 48 h old. A greater number of cells were collected from participants brushed with the Endoscan brush when under GA compared to those who underwent nasal brushing with the Endoscan brush while awake. We anticipate that this is because when participants are under GA, the operator can brush more extensively, and as a result, collect more cells.

In our study, our awake cohort were older than the GA group. This results from the standard practice in the CF clinic of routine bronchoscopy in the under-five-year-olds. Therefore, participants in phase two of our study who were under five years of age were not approached to undergo an awake nasal brush. We are therefore unable to assess the feasibility of using the Endoscan brush in this age group when awake. Additionally, we should note that participants below six months were not tested as part of our study, and as such, we cannot conclude the comparative efficiency of the Olympus and the Endoscan brush in the 0–6 months old cohort. It is worth noting that all participants included in this study had a diagnosis of CF. Therefore, while our findings may not be generalised to other conditions, we do not have any evidence to suggest that the performance of the brushes will be altered in other airway conditions such as PCD. More importantly, for PCD patients who may present with symptoms below six months of age, the investigation of this younger cohort would be important for future studies.

A previous study has compared an interdental brush with a curette and swab in adult participants [19]. Stokes et al. found that the interdental brush collected the most HNE cells with a similar cell viability across all three instruments tested [19]. They used one instrument per patient and brushed each nostril twice with a total of four of the same instrument type. In contrast, our study compared different types of brushes within the same individual, controlling for any variability between different individuals. When comparing our results, the Endoscan brush collected more HNE cells in our awake cohort than the amount obtained via the interdental brush used in Stokes’s study (14 × 10^5^ vs. 9.8 × 10^5^) [19]. Hussain et al. also collected fewer cells (1.57 × 10^5^) with an interdental brush after local anaesthetic was sprayed on the nasal mucosa of adult participants [24]. In comparison, a single Endoscan brush in our study collected 14-fold more cells in participants under GA than the amount reported for the interdental brush using local anaesthesia (22 × 10^5^ vs. 1.57 × 10^5^) [24]. Various sizes of Olympus cytology brushes are available, with diameters ranging between 2 mm to 5 mm. We did not identify any published studies comparing nasal epithelial samples collected using two different sizes of the same cytology brush. However, studies comparing different brushes for collecting bronchial epithelial cells demonstrated that while the size of the brush did not affect the number of cells obtained, the size of the bristles did [25]. However, these brushes all had smaller diameters than the Endoscan brush in our study (1.0 mm, 1.73 mm, 3 mm and 5 mm) [25]. Our study compares two brushes of different sizes, which are also different shapes and sourced from different manufacturers. Future studies may choose to compare two brushes that differ only in diameter to investigate the effect of brush width on nasal cell yield. This could help elucidate whether the diameter of the cytology brush alone impacts cell yield or whether other factors, such as shape or source, also play a role.

Various nasal brushing techniques have been described and are dependent on the individual operator, which may impact outcomes. The brushing method by Hussain et al. describes the mid-section of the inferior nasal turbinates being scraped using an interdental brush under direct vision using a nasal speculum [24]. Park et al. describe using a customised nasal brush, inserting it between the inferior turbinate and nasal septum and rotating it 15–20 times [9]. In our protocol, the brushing procedure limited the back-and-forth movement of the brush to the insertion and removal and obtained the cells by rotating the brush gently. Overall, the Endoscan brush appears to be more efficient in collecting HNE cells than previously reported cytology brushes.

Whilst high cell counts are important for successful cell culture initiation, a lack of microbial contamination is also key to the successful reprogramming and expansion of HNE cells since microbial organisms can overrun a culture and are detrimental to cell growth and viability. This study was conducted in a cohort of patients participating in a CF research study. As such, all the participants in the retrospective audit had a CF diagnosis. One key aspect of CF pathophysiology is a high microbial load in the airways (nasal passages included) [26]. Patients with an active colonisation or infection of the nasal airway with microbes resistant to our standard culture antibiotics were at risk of failure, regardless of cell counts obtained. Despite this, only 9% of participants were unable to be biobanked due to the microbial contamination of their cell cultures.

Health economics plays a pivotal role in making informed decisions about the allocation of resources and improving healthcare outcomes. With a price difference of over AUD 42 per brush (Endoscan: <AUD 0.50 vs. Olympus: AUD 42.50), the Endoscan brush is a significantly more cost-effective option. Its softer bristles and larger surface area provide a more efficient cell collection methodology, as demonstrated by the significantly higher total and live cell count it collected in this study compared to the alternative Olympus brush. Importantly, we did not demonstrate a difference in CBF measurements between the two brushes, indicating that the Endoscan brush did not compromise on diagnostic quality. It is important to note that while our study focused on CBF, the assessment of the ciliary beat pattern (CBP) is also crucial in the diagnosis of PCD [6,27] and may require a more extensive examination of the sample to identify subtle motility defects. Differences in the quality of the sample could potentially affect the quality of CBP assessment [25], and this aspect should be taken into consideration in future studies. A high cell yield may decrease the time required to identify viable nasal sheets to conduct the high-speed video microscopy assessment of CBF and CBP. In addition, sufficient material will be provided for immunofluorescence and electron microscopy assessment. This may reduce the labour costs involved in PCD diagnostic testing and could improve accuracy by providing the option of the analysis of additional patient cells.

It is important to note that our study did not assess pain and discomfort between the two brushes, as all participants were under GA during the brushing procedure. Future studies might explore this aspect, as well as investigate the impact of operator-dependent nasal brushing techniques on the outcomes, in order to further optimize the process of collecting nasal epithelial cells for various clinical and research purposes.

## 5. Conclusions

In conclusion, our study provides valuable insights into the effectiveness and efficiency of two different cytology brushes used for collecting nasal epithelial cells in the context of diagnosis and cell culture techniques. The results indicate that the Endoscan brush is not only suitable for use in paediatric patients as young as one year of age who are under GA but is also able to collect a significantly higher number of live cells compared to the Olympus brush. Importantly, no difference in CBF measurement was observed between the cell samples obtained using the two brushes, suggesting that the choice of brush does not affect the diagnostic accuracy of CBF measurement.

Additionally, our findings highlight the cost-effectiveness of the Endoscan cytology brush, which is considerably less expensive than the Olympus brush, without compromising the quality of the samples collected. This could potentially lead to more efficient resource allocation in healthcare settings, ultimately improving patient outcomes.

## Figures and Tables

**Figure 1 jpm-13-00864-f001:**
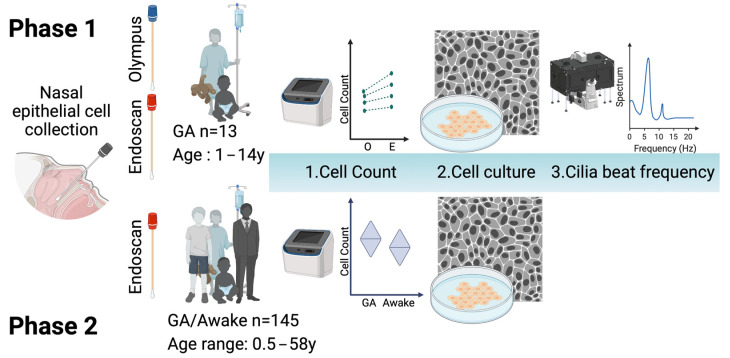
Schematic of study design. The study consists of two phases. In phase one, 13 paediatric participants with CF, aged between 1–14 years, underwent paired nasal brushing with two different brushes (Olympus and Endoscan) while under general anaesthesia (GA). The Olympus and Endoscan brushes were used on the opposite inferior nasal turbinates of each participant. The assessment included (1) total and live cell counts, (2) the morphology of cultured cells and (3) cilia beat frequency. In phase two, a retrospective audit of the Endoscan brush used in 145 participants was conducted. Nasal brushing under general anaesthesia and in an awake state were compared in order to establish the difference in nasal brush cell counts and the successful initiation of cell culture across a large age range of six months to 58 years (y). Figure created with BioRender.com.

**Figure 2 jpm-13-00864-f002:**
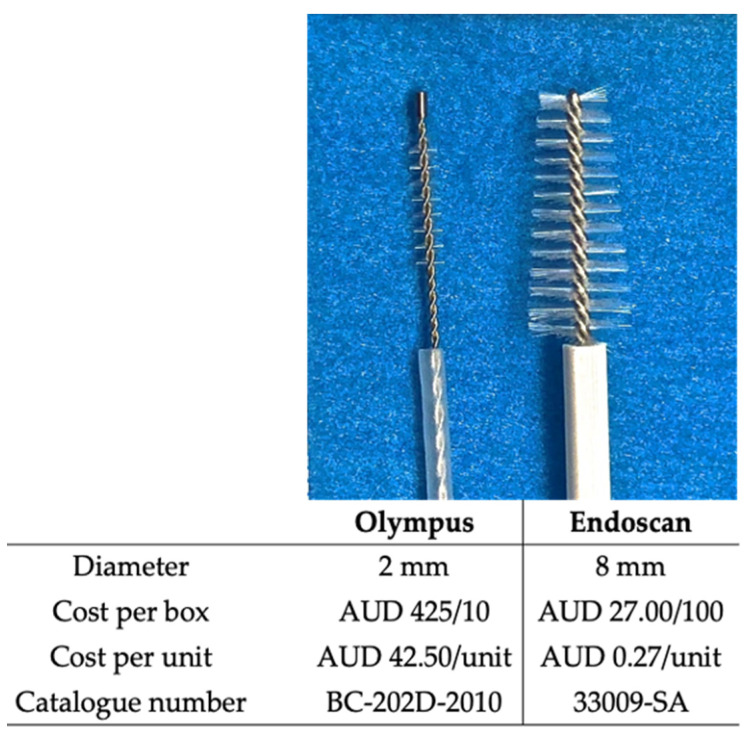
Olympus (**left**) and Endoscan (**right**) cytology brushes. The largest diameter of each brush, 2019/2020 cost and product catalogue number is recorded alongside each respective instrument.

**Figure 3 jpm-13-00864-f003:**
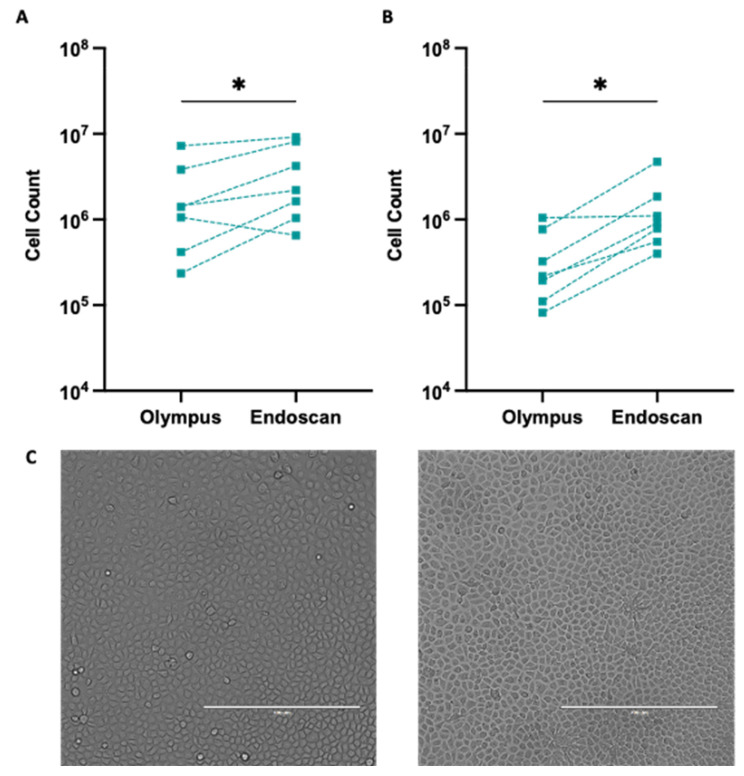
Cell counts and cultures from epithelial cells obtained with either the Olympus or Endoscan brush. Total (**A**) and live (**B**) cell counts obtained using each brush in seven participants. Wilcoxon’s matched pairs signed-rank test was used to compare the cell counts obtained by each brush. Data were log transformed prior to plotting for clarity of visualisation. Paired samples are connected using a dashed line, *n* = 7, * = *p* < 0.05. (**C**) Brightfield light microscopy showing the typical cobblestone morphology of epithelial cells at confluency. Cultures created from Olympus (**left**) and Endoscan (**right**) brush. Scale bars: 400 μm.

**Figure 4 jpm-13-00864-f004:**
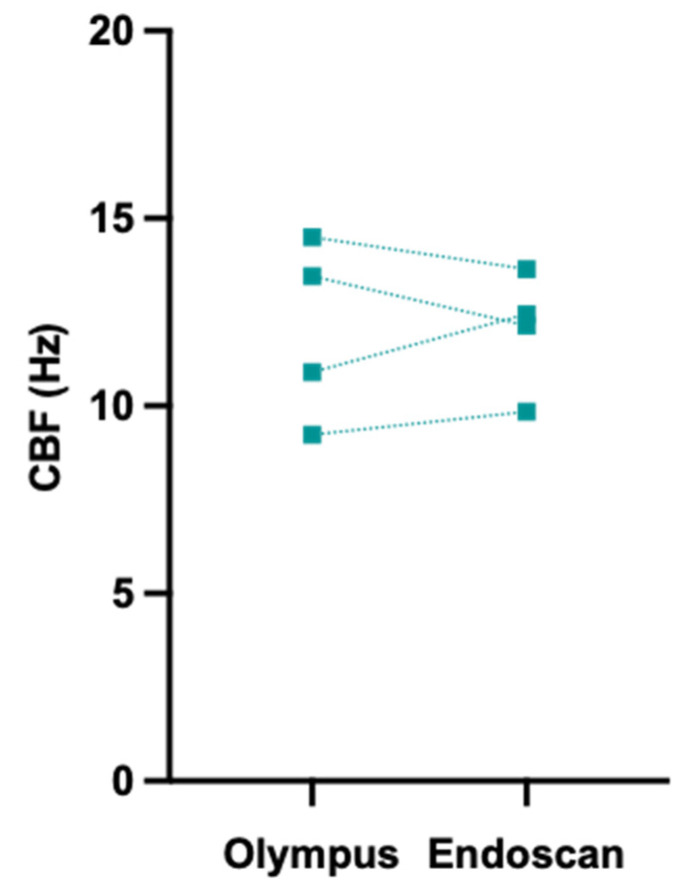
Cilia beat frequency (CBF) for nasal epithelial sheets collected with the Olympus and Endoscan brushes. Mean CBF measurements from nasal epithelial sheets obtained using both brushes for the same participant. Paired samples are connected using a dashed line, *n* = 4 participants.

**Figure 5 jpm-13-00864-f005:**
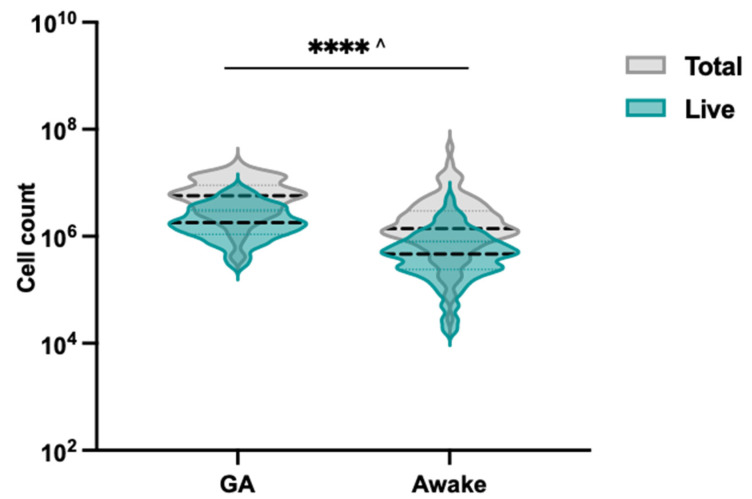
Total and live cell counts obtained from 145 participants with CF who underwent the brushing of the inferior nasal turbinates with the Endoscan brush. Data were log-transformed prior to plotting as violin plots and conducting statistical analysis. A *t*-test was used to compare the awake group to the anaesthetised (GA) group *****p* < 0.0001, ^ indicates significance for both total and live cell comparisons.

**Table 1 jpm-13-00864-t001:** Demographics of paediatric participants brushed under general anaesthetic (GA) for comparative assessment of the Olympus and Endoscan brushes in phase one of the study.

	Participants
*n*	13
Male (%)	85
Median Age (years)	3.8
Age Range (years)	1.0–14.2
Sedation state	GA

**Table 2 jpm-13-00864-t002:** Demographics of participants brushed with the Endoscan brush under general anaesthetic (GA) or in the awake state in phase two of the study.

	GA	Awake
*n*	52	93
Male (%)	51	46
Median Age (years)	4.0	15.5
Age Range (years)	0.6–17.8	5.7–58.5

## Data Availability

All data is contained within the article or Appendix A.

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
