# Peer review of "Comparing Cytology Brushes for Optimal Human Nasal Epithelial Cell Collection: Implications for Airway Disease Diagnosis and Research"

_jpm, 2023, doi:10.3390/jpm13050864_

Round 1
Reviewer 1 Report
The study is well described overall and provides some useful insights about diagnosis of airway diseases, especially primary ciliary dyskinesia (PCD). The article helps in informing the discussion about the cost-effectiveness of PCD diagnosis and the further standardization of diagnostic and research methods across different countries. Especially in countries with limited resources.
Abstract: I would add the diameter of the brushes directly in the abstract as this is important information for the reader.
Introduction: The authors state that HNE are particularly useful in diagnosing PCD and other airway diseases. To my knowledge, these are useful only for diagnosing PCD, not other diseases. Also references 4-7 all refer only to PCD, and only reference 8 is about Cystic Fibrosis, and surely brushing of nasal cells is not used in the diagnosis of CF. The authors can state are used for research or other purposes in the case of diseases other than PCD. Please rephrase.
Overall methods are clearly described as well as results. However, I believe certain aspects were not adequately addressed and several limitations exist that reduce the generalizability and impact of this work. My major comments are provided below:
Major comment 1:
The foremost population that the results of this study apply (patients suspected of PCD) is not included. Authors could discuss this limitation and explain if there are any particular constraints regarding the generalizability of findings and the use of these cytology brushes to this population. Also note, that the youngest pediatric subject in the study was 1 year old (based on table 1) or 6 months old (based on what is mentioned in the discussion). Note that in countries with good awareness for PCD, infants suspected of PCD may be identified from birth and referred for diagnostic testing soon after. So, biopsies may take place on infants aged only 1 or 2 months.
Major comment 2:
According to the authors the subjects were anaesthetized in phase 1. Again, this is not a typical case in PCD referral centers where almost all brushings take place without anesthesia. However, although the comparison in phase 2 of the study regarding the yield of cells between subjects under GA and awake subjects does confirm that the yield is acceptable when the subject is awake, the age distribution of the participants in the awake group is much older. The authors should discuss this limitation in the discussion. Why this difference in the age distribution between the two groups exists in the first place and what it could mean for the results of the study.
Major comment 3:
Based on personal and empirical experience using an Olympus 3mm diameter cytology brush results in much better cell yield for PCD diagnosis compared to the Olympus 2mm diameter cytology brush used in this study. I am not aware of any literature for this particular difference, and I believe the authors should address this point by checking if there are any published evidence about the effectiveness of the diameter of Olympus cytology brushes. A choice of an Olympus cytology brush with a different diameter may have resulted in different conclusions for this study. Even if there is no evidence, the authors should discuss this as a limitation in the discussion section.
Major comment 4:
Another point is that is unclear whether CBF determination was carried out in fresh epithelial sheets or following cell culture. Please clarify. Note that in several centers where cell culture is not an option or is not part of the PCD diagnostic pathway, CBF will only be investigated in freshly obtained cells from nasal brushings. In addition, the most important part of the HSVM analysis, is not the CBF but rather ciliary beat pattern (CBP). Although assessment of CBP is event less standardized compared to CBF (which is now largely relied on software) across the different PCD referral centers, it often requires assessment of more epithelial strips and careful examination of the sample, especially to identify subtle motility defects. I believe differences in the quality of the sample could also affect the quality of CBP assessment and this aspect was not addressed by the study. I believe this limitation should be discussed as well.
Major comment 5:
The last point I want to make is about health economics. To my understanding there are some, but overall limited evidence on the cost effectiveness of PCD diagnostic testing and this study could better inform this discussion. Please expand in the discussion section taking into consideration, not only the cost of the brush but also the implications of good cell yield for the time required and the accuracy of HSVM and electron microscopy analysis.
Author Response
Reviewer 1
The study is well described overall and provides some useful insights about diagnosis of airway diseases, especially primary ciliary dyskinesia (PCD). The article helps in informing the discussion about the cost-effectiveness of PCD diagnosis and the further standardization of diagnostic and research methods across different countries. Especially in countries with limited resources.
Abstract: I would add the diameter of the brushes directly in the abstract as this is important information for the reader.
Response: Thank you for this feedback. We have added the diameter after the name of the brush in the abstract.
This study compares the efficiency of two cytology brushes (Olympus (2mm diameter) and Endoscan (8mm diameter)) in collecting HNE cells.
Introduction: The authors state that HNE are particularly useful in diagnosing PCD and other airway diseases. To my knowledge, these are useful only for diagnosing PCD, not other diseases. Also references 4-7 all refer only to PCD, and only reference 8 is about Cystic Fibrosis, and surely brushing of nasal cells is not used in the diagnosis of CF. The authors can state are used for research or other purposes in the case of diseases other than PCD. Please rephrase.
Response: Thank you for identifying this misleading sentence. We have rephrased it and added the references for asthma and chronic rhinosinusitis studies to this section.
The procedure collects human nasal epithelial (HNE) cells from the inferior nasal turbinate, which are particularly useful in diagnosing primary ciliary dyskinesia (PCD) and are increasingly used in research of other airway diseases [2,4–9].
Vanders, R.L.; Hsu, A.; Gibson, P.G.; Murphy, V.E.; Wark, P.A.B. Nasal Epithelial Cells to Assess in Vitro Immune Responses to Respiratory Virus Infection in Pregnant Women with Asthma. Respir Res 2019, 20, 1–6, doi:10.1186/S12931-019-1225-5
Park, D.Y.; Kim, S.; Kim, C.H.; Yoon, J.H.; Kim, H.J. Alternative Method for Primary Nasal Epithelial Cell Culture Using Intranasal Brushing and Feasibility for the Study of Epithelial Functions in Allergic Rhinitis. Allergy Asthma Immunol Res 2016, 8, 69, doi:10.4168/AAIR.2016.8.1.69.
Overall methods are clearly described as well as results. However, I believe certain aspects were not adequately addressed and several limitations exist that reduce the generalizability and impact of this work. My major comments are provided below:
Major comment 1:
The foremost population that the results of this study apply (patients suspected of PCD) is not included. Authors could discuss this limitation and explain if there are any particular constraints regarding the generalizability of findings and the use of these cytology brushes to this population. Also note, that the youngest pediatric subject in the study was 1 year old (based on table 1) or 6 months old (based on what is mentioned in the discussion). Note that in countries with good awareness for PCD, infants suspected of PCD may be identified from birth and referred for diagnostic testing soon after. So, biopsies may take place on infants aged only 1 or 2 months.
Response: Thank you for this comment. We have incorporated an additional paragraph on limitations in the discussion of our manuscript and have addressed your concern accordingly.
Specifically we now state:
“Additionally, we note participants below six months were not tested as part of our study, as such we cannot conclude the comparative efficiency of the Olympus and the endoscan brush in the 0 - 6 months old cohort.”
Furthermore we have added
“It is worth noting that all participants included in this study had a diagnosis of CF. Therefore, while our findings may not be generalised to other conditions, we do not have any evidence to suggest that the performance of the brushes will be altered in other airway conditions such as PCD.”
Major comment 2:
According to the authors the subjects were anaesthetized in phase 1. Again, this is not a typical case in PCD referral centers where almost all brushings take place without anesthesia. However, although the comparison in phase 2 of the study regarding the yield of cells between subjects under GA and awake subjects does confirm that the yield is acceptable when the subject is awake, the age distribution of the participants in the awake group is much older. The authors should discuss this limitation in the discussion. Why this difference in the age distribution between the two groups exists in the first place and what it could mean for the results of the study.
Response: Thank you, we have clarified that the study cohort are patients with CF and have expanded on the limitation section in our discussion. The reason for the difference in age distribution in our study is as a result of;
- Standard practice in our CF clinic to perform annual clinical bronchoscopy for patients between age 0-5years. This provided us with an opportunity to compare the two brushes while the participant is under GA and to minimise participant reaction.
- Since it is not standard practice for patients with CF over the age of 5 years to undergo a bronchoscopy the only opportunity for participation in phase 2 was at an outpatient clinic and they therefore were brushed when awake.
- In cases where a participant above the age of 5 was scheduled for a procedure under GA, the research nasal brushing was taken at the time of the GA to minimise participants’ discomfort.
We have added a paragraph to the discussion which includes:
“In our study our awake cohort were older than the GA group. This results from the standard practice in the CF clinic of routine bronchoscopy in the under-five’s. Therefore, participants in phase two of our study who were under five years of age were not approached to undergo an awake nasal brush. We are therefore unable to assess the feasibility of using the endoscan brush in this age group when awake”.
Major comment 3:
Based on personal and empirical experience using an Olympus 3mm diameter cytology brush results in much better cell yield for PCD diagnosis compared to the Olympus 2mm diameter cytology brush used in this study. I am not aware of any literature for this particular difference, and I believe the authors should address this point by checking if there are any published evidence about the effectiveness of the diameter of Olympus cytology brushes. A choice of an Olympus cytology brush with a different diameter may have resulted in different conclusions for this study. Even if there is no evidence, the authors should discuss this as a limitation in the discussion section.
Response: Thank you for your comment. We appreciate your personal and empirical experience with the Olympus 3mm diameter cytology brush and your suggestion to investigate the effectiveness of different diameters. Unfortunately, we could not find any published studies comparing two cytology brushes of any type for nasal epithelial cells, only one study comparing an interdental brush against other instruments. However, we agree that a study where the only difference between brushes was the diameter would allow for better conclusions as to whether it is the diameter alone that impacted cell yield. We did however find reference to a 1987 and 1975 study regarding bronchial brushings. We have added a line in the discussion section acknowledging the availability of various size Olympus cytology brushes. We also mentioned that future studies may choose to compare two brushes that differ only in diameter to investigate only the effect of brush width on cell yield.
We have added;
“Various sizes of Olympus cytology brushes are available, with diameters ranging between 2mm to 5mm. We did not identify any published studies comparing nasal epithelial samples collected using two different sizes of the same cytology brush. However, studies comparing different brushes for collecting bronchial epithelial cells demonstrated that while the size of the brush did not affect the number of cells obtained, the size of the bristles did[25]. However, these brushes were all smaller diameters than the endoscan brush in our study (1.0mm, 1.73mm, 3mm, 5mm) [25]. Our study compares two brushes of different sizes, which are also different shapes and sourced from different manufacturers. Future studies may choose to compare two brushes that differ only in diameter to investigate the effect of brush width on nasal cell yield. This could help elucidate whether the diameter of the cytology brush alone impacts cell yield or whether other factors, such as shape or source, also play a role."
Major comment 4:
Another point is that is unclear whether CBF determination was carried out in fresh epithelial sheets or following cell culture. Please clarify. Note that in several centers where cell culture is not an option or is not part of the PCD diagnostic pathway, CBF will only be investigated in freshly obtained cells from nasal brushings. In addition, the most important part of the HSVM analysis, is not the CBF but rather ciliary beat pattern (CBP). Although assessment of CBP is event less standardized compared to CBF (which is now largely relied on software) across the different PCD referral centers, it often requires assessment of more epithelial strips and careful examination of the sample, especially to identify subtle motility defects. I believe differences in the quality of the sample could also affect the quality of CBP assessment and this aspect was not addressed by the study. I believe this limitation should be discussed as well.
Response: Thank you for your comment and for raising important points regarding the ciliary function analysis in our study. To clarify, CBF determination were carried out on freshly collected nasal epithelium sheets and not on cultured cells.
We have edited the following sections for clarity in the methods section 2.6;
“Briefly, collected nasal epithelial sheets were imaged using a high-speed live cell imaging system (Eclipse Ti2-E, Nikon microscope with an Andor Zyla 4.2 sCMOS camera and a CFI S Plan Fluor ELWD 20×/0.45 objective) in an environmentally controlled chamber (37°C, 85% humidity and 5% CO2) on the same day they were acquired from the participant [20].”
And results section 3.2;
“CBF measurements for freshly brushed (as opposed to cultured) nasal epithelial cell sheets acquired using both brushes were compared”.
Regarding your point about the assessment of ciliary beat pattern (CBP), we agree that this is an important aspect of PCD diagnosis. We have discussed this point by adding the following lines to our discussion:
“It is important to note that while our study focused on CBF, the assessment of ciliary beat pattern (CBP) is also crucial in diagnosis of PCD [6,27] and may require more extensive examination of the sample to identify subtle motility defects. Differences in the quality of the sample could potentially affect the quality of CBP assessment [25], and this aspect should be taken into consideration in future studies.”
Major comment 5:
The last point I want to make is about health economics. To my understanding there are some, but overall limited evidence on the cost effectiveness of PCD diagnostic testing and this study could better inform this discussion. Please expand in the discussion section taking into consideration, not only the cost of the brush but also the implications of good cell yield for the time required and the accuracy of HSVM and electron microscopy analysis.
Response: Thank you for this comment. We agree and have expanded the health economics paragraph in the discussion by adding the following at line;
“A high cell yield may decrease the time required to identify viable nasal sheets to conduct high speed video microscopy assessment of CBF and CBP. In addition, sufficient material will be provided for immunofluorescence and electron microscopy assessment. This may reduce the labour costs involved in PCD diagnostic testing and could improve accuracy, by providing the option of analysis of additional patient cells.”
Reviewer 2 Report
To the Editor,
In their interesting study, Laura K Fawcett and Shafagh A Waters et al. compared efficiency of two types of cytology brushes in the sampling of nasal epithelial cells (HNE) from pediatric cystic fibrosis patients (n=13, sampled under general anesthesia). The Authors show that the 8mm Endoscan brush (cheap brush used routinely for endocervical cytology) sampled more viable cells compared to the 2mm Olympus brush (a standard for sampling of bronchial cells during bronchoscopy). The cilia beat frequency measurements were comparable regardless of the sampling method. Both methods were efficient in recovering HNE cells for in vitro culture. Notably, to allow rapid expansion of epithelial cells, they used recently introduced ‘dual-SMAD inhibition’ cell culture method. In the second part, the Authors confirm that Endoscan brush is also useful in cell sampling from awake patients (n=145; wide age range) with a nearly 90% success rate of HNE cell culture. The high efficacy of sampling is critical for the recovery of airway cells for in vitro research. Therefore, this methodological study is important to researchers dealing with nasal sampling to establish biobanks of airway cells and for diagnostic purposes, especially in younger children. The cost effectiveness of using a very inexpensive cytology brush is also worth mentioning. The study is clear and concise, and I have no significant critical remarks.
Minor.
1. Please check reference in line 142. In this paper, Wong et al (Ref 13) actually used CR cell culture method for cell expansion.
2. If applicable, could the Authors provide more data on cell expansion capacity of sampled cells. Currently, only side-by-side images (Fig.3C) are shown. For the second part, are the initial cell yield and viability important for cell expansion (eg, doubling rate etc). But these are just my suggestions, not a requirement to provide these data.
3. A perfect complement to this paper could be a short video (images?) showing the optimal method of collecting cells using Endoscan brushing from an awake person (actually, one of the key messages of this study).
Round 2
Reviewer 1 Report
Thank you for addressing my concerns. No further comments.